Central African Republic; group intervention; conflict and war; mental health; trauma

**Corresponding author:**
William John Froming;
Email: wfroming@paloaltou.edu

# RCT of post-conflict trauma interventions in the Central African Republic

William John Froming[1] , Maryke Van Zyl[2], Karen Bronk Froming[1],
Vicky Bouche[1,3] and Sita G. Patel[1]

[1]Department of Psychology, Palo Alto University, Palo Alto, CA, USA; [2]Department of Psychiatry, San Francisco Veterans Health Administration, San Francisco, CA, USA and [3]Department of Psychology, Rady's Children's Health, Orange, CA, USA

## Abstract

This study evaluates mental health treatment in a post-conflict setting with scant mental health resources. The study reports on a randomized crossover control group design with one intervention and two control groups implemented in the Central African Republic (CAR).

The intervention's impact on symptoms of depression, anxiety and trauma was analyzed among a sample of 298 participants located in the capital city, Bangui. Participants were screened for elevated levels of anxiety and depression and randomly assigned to one of three groups: control, intervention and active control. Data included an initial interview, measurement following the two intervention workshops and a 3-month post-intervention follow-up.

The trauma reduction intervention significantly reduced symptoms of depression, anxiety and trauma compared to the waitlist control. The active control group focused on peace and value education and produced equivalent outcomes to the trauma-reduction intervention group. Further, at 3 months follow-up, the impact of both interventions remained significant, although lower. The two interventions did not differ from one another.

The study demonstrates two practical approaches for addressing anxiety, depression and trauma symptoms in post-conflict, low-resource settings. The similar outcome of the two interventions may suggest that they share common therapeutic elements.

## Impact statement

The Central African Republic (CAR), one of the most resource-poor countries in the world, has been ravaged by armed conflict for decades. CAR is considered a low- and middle-income country (LMIC; World Health Organization [WHO], 2017), and its people continue to suffer due to significant gaps in mental health treatment. The immense mental health gap in LMIC worldwide is, in part, related to a scarcity of effective programs that address the mental health burden of armed conflict in the civilian population (WHO, 2017). The WHO (2015) has provided suggestions for bridging mental health gaps in LMIC, such as CAR. These suggestions include implementing evidence-based programs through task shifting or task sharing, which involves interventions delivered by nonspecialists.

This study examined the impact of a psychosocial intervention that addressed trauma and compared it to an active control group that emphasized peace and value education and a waitlist control. The trauma-focused intervention, as well as the active control group, significantly reduced mental health symptoms compared to the waitlist control group, suggesting that these interventions may share common factors (Pedersen et al., 2020). In addition, these empirically supported mental health interventions were delivered by nonspecialists from the region, providing support for task shifting (WHO, 2022) to address gaps in treatment.

These interventions offer a way to reduce the levels of distress among communities in post-conflict societies, thereby increasing the capacity to reengage with their families and their economic tasks, helping to restore a sense of normalcy. Beyond the impact of these interventions on the people of CAR, this work supports the task-shifting model as well as immediate brief psychosocial interventions.

## Introduction

One in five people who have experienced armed conflict develops a mental health diagnosis within 10 years, which may include depression, anxiety, post-traumatic stress disorder (PTSD), bipolar disorder or schizophrenia (Charlson et al., 2019). Other symptoms experienced when exposed to armed conflict include social isolation, emotional distress, insomnia, family conflict and a tendency toward aggressive or risky behavior (Van der Kolk et al., 1995; Betancourt et al., 2009;

Dorrepaal et al., 2010). Post-conflict settings present a complex, interactive milieu of social and structural determinants of health inequalities, including mental health, involving most aspects of day-to-day living. Furthermore, the most vulnerable members of the population are women, children, internally displaced persons and people with preexisting symptoms of mental illness (Bwirire et al., 2022).

In addition to the mental health burden on individuals, family systems are significantly impacted by armed conflict through the loss of lives (Weine et al., 2004), family separation and displacement (Bass et al., 2013) and financial strain (Patel, 2012). Families are often displaced and separated during armed conflict and experience severe losses in social and financial support, leaving survivors to take on multiple caregiving roles on behalf of those who were displaced, left behind, injured or killed. The effects of trauma on families can be long-standing and intergenerational, leading to ripple effects that negatively impact communities long after a conflict has ended (Pearrow and Cosgrove, 2009; Bezo and Maggi, 2015; Sangalang and Vang, 2017). Where neighbors with different ethnic or religious backgrounds may have coexisted or even coalesced in a pre-conflict society, communities are often fractured due to public misinformation, politically charged bias and interethnic conflict (Ajdukovic and Biruski, 2008). The cycles of conflict-inflicted poverty and trauma have been well-documented (Lund et al., 2011; Patel, 2012; Chisholm et al., 2016; Patel et al., 2018) regarding broad community burdens.

### Violent conflict in the CAR

The current study was conducted in the CAR. CAR has many ethnic groups (e.g., Twa, Gbaya, Banda, Mandjia, Sara, Mbum and Ngbaka ethnic groups) and an extensive history of interethnic, interreligious and colonial conflicts (Carayannis, 2015; Lombard, 2016). CAR has experienced decades of political instability and violence, including six governmental coups (Gang et al., 2023). In a recent outbreak of violence, starting in 2013, an alliance of predominantly Muslim militia groups named Seleka overthrew the government and seized power. An alliance of predominantly Christian militia groups, known as the Anti-balaka, responded and drove the Seleka out of the capital city of Bangui. During this period, ~1.2 million people were displaced, either internally or to surrounding countries, and several thousand were killed (United Nations High Commissioner for Refugees [UNHCR], 2024). More recently, the UN peacekeeping forces have increased, and ongoing violence still occurs regularly (Global Centre for the Responsibility to Protect [GCR2P], 2025; UNHCR, 2024; World Bank, 2024).

Unfortunately, CAR lacks the mental health infrastructure to manage the psychological consequences of the conflict. CAR has almost none of its own healthcare (Baleta, 2021; United Nations Office for the Coordination of Humanitarian Affairs, 2024) or mental health resources (Vinck and Pham, 2010), with only one practicing physician with psychiatric training in a country of 4.6 million people at the time of this study (Baxter and Allison, 2020). Current population estimates suggest an increase to over 6 million people, with some psychosocial workers providing services to only 2,982 people in 5 district health centers (World Bank, 2024). Mental health resources are primarily provided by nongovernmental organizations (NGOs; Bass et al., 2006), which means that providers are typically trained in the United States or Europe, work primarily in the capital and tend to only stay for a few months (Baxter and Allison, 2020), leaving the community in flux. This model of mental healthcare

is unsustainable, provides only temporary care and does not utilize local resources.

To provide quality global mental health care for all, a few key strategies have enabled more people to access appropriate care (Patel and Thornicroft, 2009; WHO, 2019). These strategies include thorough needs assessments (Patel, 2012; Becker and Kleinman, 2013), adapting Eurocentric approaches to the relevant context (Bolton et al., 2003; Patel et al., 2020) and involving local stakeholders, such as community members and religious leaders, through task shifting (Patel et al., 2020). Before implementing mental health programs, a thorough understanding of the population, possible barriers and common challenges experienced by similar organizations (Patel, 2012; Becker and Kleinman, 2013) is necessary.

### The present study

The conflict in CAR, which started in 2012–2013 between the Seleka and the anti-Balaka, came to the attention of the UN Security Council. A Commission (The International Commission of Inquiry on the Central African Republic) was appointed and a report was filed on the situation (United Nations Security Council, 2014). The report noted the death and destruction already present, and that CAR was vulnerable to additional violence. While it did find genocidal acts had occurred, it did not find evidence of genocidal intent on the part of any of the combatant groups but did not rule it out for the future.

The situation continues to be dire with violence and insecurity erupting periodically. It was recently estimated that about 60% of the population is traumatized (Dozio et al., 2021). In response to the complex and extreme humanitarian crisis, the US Agency for International Development developed a multipronged plan for improving the CAR situation: (1) promote civil institutions in establishing a leadership role in peacebuilding, (2) reestablish livelihood security and (3) foster social cohesion through psychological intervention. The plan was carried out by a number of NGOs that were already active in CAR: Catholic Relief Services (CRS), Aegis Trust (AT), Islamic Relief Worldwide (IRW) and World Vision International (WVI).

The focus of this study centers on the effort to improve mental health conditions in the traumatized population (the third of the goals outlined above). Identifying empirically supported treatment(s) for improving mental health during quiet periods or in post-conflict situations can improve the lives of individuals and their social networks. The current study rigorously tested the efficacy of a trauma reduction intervention that had not been subjected to extensive testing. Two comparison groups were included. The study utilized cross-culturally validated measures, independent data gatherers and a randomized crossover design. We build on our earlier work (Patel et al., 2020), which not only sheds light on the dire need for mental health services in CAR but also provides vital information to guide culturally congruent interventions. Findings from that study confirmed the prevalence of traumatic experiences and traumatic stress among survivors of armed conflict in this region, as also found in previous studies (Bolton, 2001; Bass et al., 2006; Vinck and Pham, 2010; Patel et al., 2020). Patel et al. (2020)) described conducting focus groups in CAR that yielded the following three themes: (1) the need to reestablish safety, (2) trauma experiences and local manifestations of distress and (3) local mental health treatment.

Two task-shifting interventions were implemented by community leaders who had been trained by CRS and Aegis staff members. The first intervention, Healing and Rebuilding Our Communities

(HROC), was explicitly intended to improve aspects of mental health by reducing high levels of anxiety, depression and trauma symptoms as culturally described by community members. Prior workshop participants reported that they found the workshop very helpful, and it has been used in several countries in sub-Saharan Africa, the United States and Canada. Yeomans et al. (2010) used two variants of HROC and a waitlist control group in Burundi to examine the efficacy of HROC. They found both versions of HROC improved trauma symptom reporting but not broader measures of stress. The remainder of available unpublished outcome data are generally limited to pre- and posttest ratings on the Harvard Trauma Questionnaire (HTQ) and the Hopkins Symptom Check-list (HSCL-25) along with informal comments (Karen B. Froming, personal communication, 2015).

The second intervention, AT's Peace and Values Education (PVE) was developed in the aftermath of the Genocide Against the Tutsi in Rwanda. Based on the work of Staub (Staub, 1994, 2015) and Stanton (1999), it utilizes an African modality of oral history and storytelling (Afriklens, 2024).

> Community is at the heart of African oral traditions. Storytelling sessions are not solitary activities but communal gatherings where people of all ages come together to engage in a shared experience. These gatherings allow communities to bond, transmit collective values, and celebrate their heritage. In African societies, storytelling is often performed in an interactive manner, with the audience responding to prompts or adding their input. This participatory element creates a dynamic atmosphere and strengthens the connection between the storyteller and the audience.

The ultimate goals of PVE were to experientially learn about empathy, tolerance, critical thinking and personal responsibility through facilitated dialogue. Although the PVE intervention has merit as a standalone workshop and has been adopted by schools throughout Rwanda, its content was not designed to improve emotional and mental health per se. Its intent is to improve critical thinking skills, empathy and social responsibility, thereby hoping to prevent people from falling prey to genocidal ideology. By comparing the impact of HROC to PVE, we hoped to better understand the critical components of the HROC intervention. The following provides a detailed description of the three experimental groups:

**1. HROC** was derived from the Quaker program "Alternatives to Violence Project" (AVP). AVP was created in 1975 for work with prisoners (John, 2021). HROC originated in Rwanda in 2003 in the aftermath of the Genocide against the Tutsi in 1994. Groups of 15–25 individuals gather for a 3-day workshop. When feasible, group composition consists of members from each side of the conflict; in CAR, that meant Christians (Catholics and Protestants) and Muslims, as well as men and women. This will be discussed in more detail in the Methods section. Day 1 of the workshop is psychoeducational and skill-building. Presentations and discussions center on differentiating between normal and traumatic stress, defining traumatic stress symptoms and their health effects. Trauma-focused self-help skills are then introduced and practiced. Day 2 involves learning active listening techniques and the discussion of loss, grief and anger through limited and confidential sharing of personal testimonies. Despite complex content, empathy is a byproduct of sharing, as violence has negatively impacted all. Day 3 focuses on mistrust and the development of trust through communal activities. More information is available on the HROC website and in the manual, which can be retrieved here: https://healingandrebuildingourcommunities.org/philosophyandapproach/. The workshops are typically led by two to three individuals who have participated in a prior workshop and then attended a 5-day training program to qualify them as leaders. They are usually psychologically minded community members with credibility (e.g., community members, religious leaders and teachers).

**2.** The **PVE** workshop was developed by AT (https://www.aegistrust.org/) in Rwanda and was culturally adapted for use in CAR. The workshop lasts 2 days and is led by two to three leaders trained by AT. The overarching goals are to develop knowledge, personal attitudes and cognitive and behavioral skills that will lead the trainees to pursue and engage in positive activities in the future. Similarly, these same factors will reduce the likelihood of the participants being influenced by opposing (i.e., genocidal) ideologies and activities in the future. The workshop seeks to "inoculate" the participants should they face conditions like those present in pre-genocide Rwanda.

The first day consists of psychoeducation about the shared colonial history of the country, discussion and exercises on cycles of violence, genocide, loss of social cohesion and witness stories. The second day focuses on the future with stories of humanity and various exercises to promote the workshop's goals. Exercises on this day explore topics, such as the road to peace, cycles of benevolence, forgiveness, critical thinking and prosocial action.

**3. Waitlist control group.** The third group in the study was a passive waitlist control group. The participants were assessed at the same time as the members of the intervention groups. They received no active treatments during the study, but participated in both treatments when the data-gathering phase concluded.

The group facilitators were experienced workshop leaders from CRS and AT. In addition, local religious and civic leaders joined in leading the workshops and provided credibility, local knowledge and translation assistance if needed.

### Comparing the two interventions

The HROC intervention focuses on the trauma experienced by the individual, for example, its origins and symptoms, while normalizing responses to extreme stressors. Participants share some traumatic experiences that invariably generate mutual empathy and commonality among group members. HROC helps participants share their anger and loss while introducing coping strategies and exercises specific to the trauma responses. Finally, the shared experience in the workshop enables all community members to begin building social cohesion. (Table 1).

The PVE program focuses on genocide (e.g., its origins with its ties to the colonial past, failed governance and inequities among members of the group/society that produce violence. The workshop draws on examples of genocides from around the world, forgiveness and ways to counteract/prevent genocide. It utilizes Socratic questioning to engage participants in discussion of shared history and didactics related to cycles of violence as well as the cycle of positive, prosocial behavior to reduce the likelihood of violence.

Thus, the content focus of the two interventions is different, as are the kinds of exercises and analyses used. However, both interventions are group interventions that aim to bring together participants from both sides of the conflict. Each intervention demonstrates and encourages active listening and interactive discussion of the material. The next phase introduces storytelling and testimonies, as well as exercises for coping. Through this phase, the identification of the commonality of experience and emotional impacts is discussed. As mentioned, PVE was not designed to improve mental health. A fair test of the PVE intervention's ability to meet its intended goals would employ different outcome measures than the ones used here. Such an analysis is beyond the scope of this article.

**Table 1.** Comparison of HROC and PVE

| HROC groups compared to PVE groups | | |
|---|---|---|
| | HROC | PVE |
| Length | 3 days | 2 days |
| Group size | 20–25 | 20–25 |
| Religious identity | Mixed | Mixed |
| Sex | Both | Both |

| HROC content compared to PVE content | | | |
|---|---|---|---|
| | HROC | | PVE |
| Day 1 | Trauma | Day 1 | Social cohesion |
| | Definition | | Common history – pre-colonial conditions |
| | Causes | | Colonial and post-colonial loss of social cohesion |
| | Symptoms | | Roots of genocide: continuum of violence |
| | Consequences | | Genocide against the Tutsi in Rwanda |
| | | | Consequences of genocide |
| Day 2 | Loss, grief and mourning | Day 2 | Recovery from genocide |
| | Dealing with anger | | Continuum of benevolence |
| | Relaxation | | Social reconstruction |
| | | | Complex nature of forgiveness |
| Day 3 | Rebuilding trust: | | Attentive listening and sharing |
| | Sources of trust | | Identifying and building critical skills |
| | Sources of mistrust | | Critical thinking |
| | Acceptance circle | | Empathy |
| | | | Active bystanders |
| | | | Personal commitment to action |

*Note*: Both workshops had activities and exercises interspersed throughout the day. The exercises were mood enhancing and/or socially interactive. They also dined together at lunch and at a tea break.

With limited empirical support in the literature for HROC as an effective mental health treatment for lowering stress-related emotions, the first hypothesis is whether HROC reduces stress more than the waitlist control group. In the context of this study, the PVE group then forms a critical comparison group for interpreting the outcomes of the HROC intervention. HROC primarily focuses on traumatic events that the participants have already experienced. PVE looks ahead as it seeks to prevent the participants from succumbing to genocidal ideology that they may be exposed to in the future. There is no data in the literature documenting that PVE impacts mental health variables, and as mentioned earlier, it was not designed to do so. Essentially, then, the PVE intervention will function as an active control condition. This study will examine that possibility. In addition, the two interventions may interact. For example, a discussion of society's role in creating the distal conditions leading up to the conflict (PVE) might lead to a greater appreciation of the proximal causes of one's personal trauma (HROC). Conversely, the HROC intervention may make participants in the PVE groups more receptive to that content.

If HROC produces significant changes in the dependent variables compared to the other two groups, all else assumed equal, the trauma focus can be assumed as the mechanism of change. If both HROC and PVE produce similar positive results (with no interaction), then we must consider whether they share anything in common that may explain their similar outcomes.

## Methods

### Recruitment and sample selection

The initial pool of participants was 1,103 adults who lived in Bangui in 2017. Religious leaders nominated many individuals for participation based on the leaders' belief that the individuals had suffered significantly because of the conflict. The senior religious leaders broadcast their joint support and advertised the work being done in an effort to quell the perception that the violence was faith-based. These leaders included the Catholic Archbishop (i.e., the senior Catholic prelate in CAR), the President of the Islamic Council (the senior Imam) and the President of CAR's Evangelical Alliance. These religious leaders provided both nominations of local religious leaders and explicit permission for them to cooperate with the project. The research team visited local churches and was introduced by the church leader, after which they answered questions from the community about the project. Using snowball sampling, these initial participants spread the word and others volunteered to be interviewed.

The HSCL-25 was chosen as the screening device for the study. Meeting the HSCL-25 cutoff of 1.75 as well as age (i.e., adults over 18 years of age) were the only inclusion criteria. It is a brief screening instrument and targets two major components of PTSD (Barbano et al., 2019). The HSCL-25 has been used in diverse cultural settings around the world and has been found to have good structural integrity (e.g., see Wind et al., 2017). While studies showed some differences, the general pattern is that the HSCL-25 factor structure replicates across these varied samples. The cutoff points can vary slightly across studies (Christy et al., 2021), but Mollica recommended 1.75, which we adopted.

The original pool of 1,103 individuals was screened using the HSCL-25, and 502 were eligible for participation because their HSCL-25 score was 1.75 or higher (Ventevogel et al., 2007; Mollica, 2012). The mean HSCL-25 score of the 298 participants in the study was 2.61, SD = 0.44. From the eligible participants, 143 were included in a pilot study to train the assessors, clear up any questions about testing procedures and identify issues with the accuracy of translation. That left a pool of 359 eligible participants (Table 2).

The remaining 359 participants were called and invited to participate in the workshops. Some people were unreachable, and some could not commit to the time requirements. The workshops were scheduled in rooms that would accommodate 20–25 participants plus workshop leaders. Typically, these rooms were in a local hotel conference room and occurred during the day. Thus, the time of day and time across multiple sessions were factors impacting who could be scheduled. The final number of participants was 298, defined as people who participated in at least three of the four assessment sessions (i.e., people who participated in the pretest, the two workshops and at least two of the three post-workshop assessment sessions). Those who were not scheduled were lost for several reasons, including being unavailable at the designated time(s), difficulty contacting them and limited numbers of workshops, among others.

**Table 2.** Sample-selection flow chart

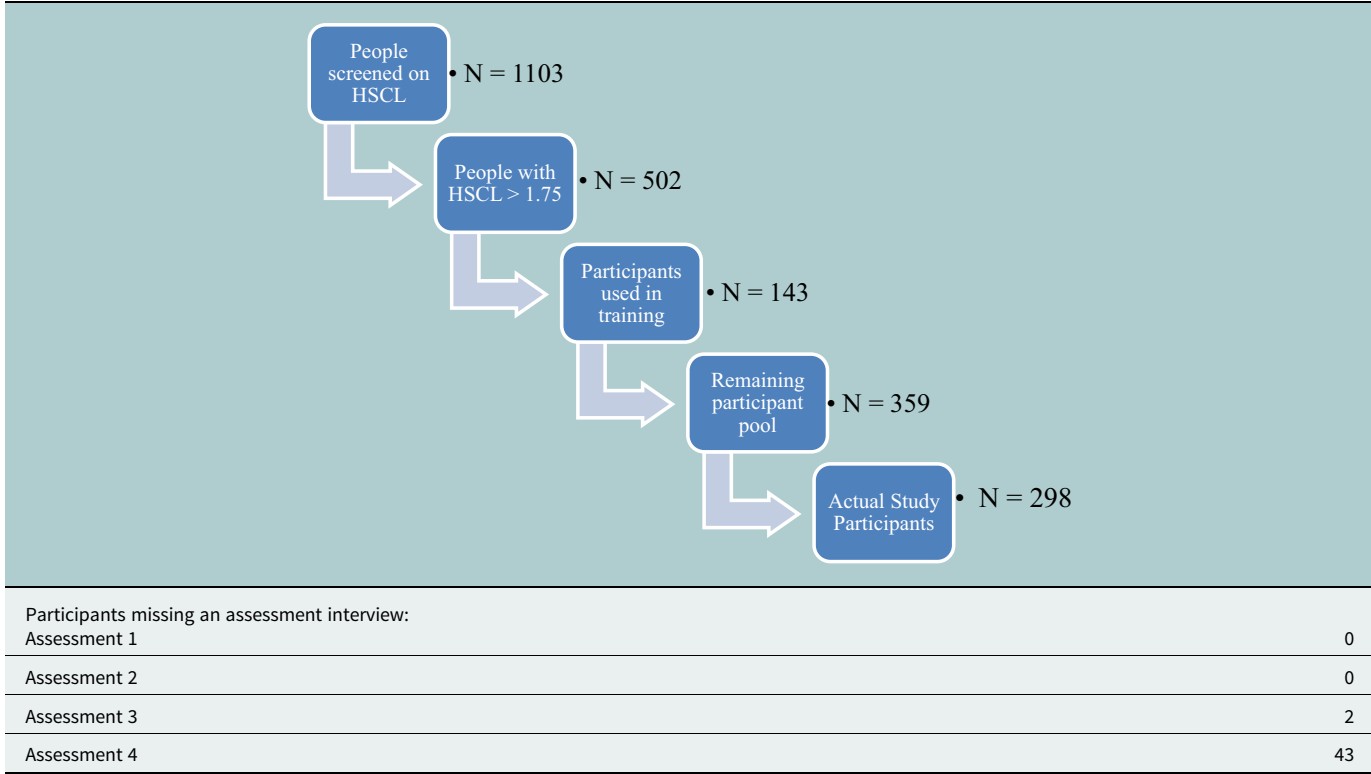

| | | |
|---|---|---|
| Participants missing an assessment interview: Assessment 1 | | 0 |
| Assessment 2 | | 0 |
| Assessment 3 | | 2 |
| Assessment 4 | | 43 |

*Note*: Other scales had missing data as well (e.g., eight participants did not respond to some questions on the HTQ).

A few (*N* = 8) participants declined to answer questions on the HTQ. We did not query them as to why they made this choice, given the personal nature of the questions, and we decided not to exclude these participants from the study as these questions were not central to our evaluation. Details of trauma experiences gathered from the HTQ (*N* = 290) are presented in Table 3.

### Sample size

Assuming a medium effect size, $\alpha$ = .05 and $\beta$ = .8, the recommended sample size was 44 participants per group or 132 in total. With the unsettled conflict environment present in CAR, we did not know how many participants would be lost over the course of the study. Therefore, we increased our sample size goals to 100 per group to guard against significant attrition.

### Participants

Participants (*N* = 298) were 70% male and 30% female, with an average age of 37.41 (SD = 12.97). The breakdown of religious affiliations was 25% Catholic, 42% Protestant and 27% Muslim. An additional 5% of the participants did not state a religious preference. It should be noted that the United States Department of State (2021) estimates the prevalence of religions in CAR as 61% Protestant, 28% Catholic, 9% Muslim and 2% other religious groups. Therefore, our sample overrepresents Muslims and underrepresents Protestants.

The single greatest factor contributing to the failure to complete the program was the 3-month follow-up. Of those who completed the program (missing no more than one assessment), 43 of 45 misses were in the follow-up session. Only two people missed a different session, but attended the follow-up session.

### Study design

Eligible participants were assigned to conditions using a random number generator.

They were randomly assigned to receive either an HROC workshop or a PVE workshop first, followed by the other workshop approximately a week later. A third group was waitlisted and assessed at the same intervals as the two treatment groups. Participants stayed in the same groups when possible. If someone missed a session, they could attend a different group to make up for it.

This resulted in a randomized crossover clinical design. Thus, participants were assessed four times throughout the process (i.e., before any workshops, after the first, after the second and 3 months later).

### Interviews and interviewers

An initial translation of the interview questions was performed by one of the team members who was trilingual. Interviewers were recruited by Echelle, a local organization recommended by other researchers in CAR. The interviewers were fluent in both Sango and French. Before conducting the interviews, the research team worked closely with interviewers to resolve any confusion over the meaning of the items as they were translated. The data questions were read in either Sango or French to the participants because literacy could not be assumed.

**Table 3.** Self-reported traumatic events on a modified version of the Harvard Trauma Questionnaire

| Trauma exposure for research participants (*N* = 290) | |
|---|---|
| Witnessed events: | *N* |
| Combat | 248 |
| Pillaging | 260 |
| Physical beatings | 253 |
| Killings | 223 |
| Sexual abuse | 105 |
| Experienced: | |
| Possessions destroyed, stolen, burned | 257 |
| Forced to flee | 269 |
| Displaced now | 86 |
| Separated from family | 250 |
| Physically beaten or attacked | 137 |
| Caught in middle of combat | 30 |
| Threatened with death | 145 |
| Expected to die | 264 |
| Friends or family members killed | 276 |
| Kidnapped: | |
| Attacked and held prisoner | 83 |
| Forced to harm someone | 41 |
| Sexual violence: | |
| Victim of sexual violence by armed group | 29 |
| Victim of sexual violence by group other than armed group | 20 |
| Forced to commit sexual violence | 91 |

A 2-day workshop was then held for members of the research team together with interviewers. One half-day of the workshop was about the use of the KoboToolbox software and computer tablets. Most of the time was spent reviewing and discussing item translation to ensure that the meaning and intent of the question were maintained. When consensus was reached on the interview questions and the interviewers were able to operate the tablets, interviews could begin. For the first set of pilot interviews, the senior member of the research team was present to oversee the data gathering and address any questions.

In the original sample of 1,103, 96% did the assessment in Sango, 2.25% did it in Arabic and the remainder did it in French or one of the other local languages. Of the 298 who were included in the study, 296 completed the assessments in Sango and two in French. The answers were input in Sango on tablets using KoboToolbox software. The data were stored numerically so they could be interpreted directly by the data analyst without further translation.

All interviews were one-on-one, lasting about 1–2 h, and were conducted in the participants' neighborhoods whenever possible. When this was not possible, the participants traveled to an office or community space that minimized distractions. In these cases, participants were reimbursed for their travel expenses. Payments were restricted to transportation costs for each session completed. Refreshments were provided during the workshops as well.

Raw answers to the 1,103 initial interviews were sent via an electronic file to the senior researcher. Participants who met the selection criteria were identified (*N* = 502). Training of the interviewers was conducted on 143 eligible participants, and thus disqualified them from the main study. Groups of 15–25 individuals were then formed using blocking factors: sex and religion (Catholic, Protestant, Muslim and other [e.g., animists]). Members of the group were called by a research team member to schedule their attendance in one of the three conditions, and 298 were successfully scheduled. Neither the people making the calls nor the participants knew to what group they were being assigned.

### Measures

The extensive interview questionnaires used in this study included instruments that were cross-culturally validated in settings around the globe and included the HSCL-25, the HTQ (Mollica et al., 2004) and the PTSD Checklist-5 (PCL-5). These instruments were translated and back-translated by team members fluent in English and Sango. Additional local experts in Sango/French translated/back-translated twice from English to Sango, and from French back to Sango. Constraints of the present study environment (funding, security situation and rapid needs for implementation) did not allow for an in-depth cross-validation of each instrument.

The HSCL-25, which was used as a screening measure, was discussed earlier in this section. For this measure, individual responses can range from 1 (*not at all*) to 4 (*extremely*) on each item. The PCL-5 measures DSM-5 symptom criteria for PTSD. The PCL-5 is a 20-item self-report measure. Responses range from 0 *(not at all)* to 4 *(extremely)* on each item. The PCL-5 has cross-cultural support from other studies (e.g., Lima et al., 2016). In considering whether to use the PCL-5 or HSCL-25, we decided to include both as they are both relevant to what we wanted to measure; however, we were looking for broad post-trauma symptom distress rather than diagnosing PTSD according to the Diagnostic and Statistical Manual-5. It should be noted that over half the sample (54%) reported that they had no education, and another 24% did not finish elementary school.

Responses to the HTQ provided evidence of the type and extent of the trauma the study participants had experienced. Over 89% of the participants reported witnessing pillaging and beatings, and 77% witnessed killings. Most (93%) of them had been forced to flee, and 91% were expected to die. Thus, the participants reported exposure to traumatic events and elevated levels of anxiety and depression, as measured by the HSCL-25.

### Institutional review

The institutional review for the study was carried out at Palo Alto University, as there was no functioning review board in Bangui. We sought consultation with an IRB in CAR regarding the proposed study, but due to the armed conflict, the University of Bangui was functionally shut down and we were unable to locate an IRB. The proposed study was discussed with a psychology professor at the University of Bangui, in an effort to approximate an in-country ethical review. The professor and some of the local NGO (CRS, AT, IRW and WVI) leaders supported the effort as its goal was to investigate treatments for widespread trauma that existed in the wake of the conflict. Dozio et al. (2021) described a similar situation.

## Results

### Preliminary analyses

#### Study participants

Scores on the HSCL-25 > 1.75 were the basis for eligibility for the study. They were randomly assigned to conditions blocked by sex and religion. The distribution within the groups by sex and religion is displayed in Table 4.

Within each experimental group, the $\chi^2$ value is not significant. Across all participants, the $\chi^2$ is significant, $p < .01$. This finding is partly due to low expected frequencies in some of the cells; however, eliminating the category of "other religions" does not eliminate the problem. It appears to be the result of larger male/female ratios among Christians and "other" religions compared to Muslims, who were roughly equally divided by sex. We were unable to more fully explain this finding.

We did not find any differences between the larger pool of qualifying participants and the actual study participants. About 17% of the qualifying pool did not participate in the actual study. Initial participation in the study was done on recommendations by clergy, imams and local leaders and done by word of mouth. Contacting potential participants to schedule them for the workshops was done by phone. While phones were/are relatively common in CAR, the living situation was very chaotic. The chaos may have introduced any number of factors that made some people easier to contact than others. The loss of 43 participants at the 3-month follow-up raised the concern that they might be different from those who participated. To address this, scores of the HSCL-25 on the *third* assessment were computed for the 43 participants who were not in the fourth assessment, compared to the other participants who were in the third and fourth assessments. This comparison was not significant, indicating that the participants leaving after the third assessment did not differ from those who went on to complete the fourth assessments.

**Table 4.** Sex and religion

| Overall | | Religion | | | | |
|---|---|---|---|---|---|---|
| | | Catholic | Protestant | Muslim | Other | Total |
| Sex | Male | 58 | 88 | 43 | 14 | 203 |
| | Female | 18 | 36 | 32 | 1 | 87 |
| Total | | 76 | 124 | 75 | 15 | 290 |
| | $\chi^2$ (3) = 11.11, $p$ = .01 | | | | | |
| | Low expected frequencies in some cells violate $\chi^2$ assumptions; if you remove the "other" category, then: | | | | | |
| | $\chi^2$ (2) = 6.86, $p$ = .032 | | | | | |
| Condition 1 | | Religion | | | | |
| | | Catholic | Protestant | Muslim | Other | Total |
| Sex | Male | 18 | 30 | 13 | 7 | 68 |
| | Female | 7 | 12 | 10 | 1 | 30 |
| Total | | 25 | 42 | 23 | 8 | 98 |
| | $\chi^2$ (3) = 3.19, ns | | | | | |
| | $\chi^2$ (2) = 1.79, ns | | | | | |
| Condition 2 | | Religion | | | | |
| | | Catholic | Protestant | Muslim | Other | Total |
| Sex | Male | 25 | 32 | 13 | 4 | 74 |
| | Female | 7 | 11 | 12 | 0 | 30 |
| Total | | 32 | 43 | 25 | 4 | 104 |
| | $\chi^2$ (3) = 7.07, $p$ = .07 | | | | | |
| | $\chi^2$ (2) = 5.26, $p$ = .07 | | | | | |
| Condition 3 | | Religion | | | | |
| | | Catholic | Protestant | Muslim | Other | Total |
| Sex | Male | 15 | 26 | 17 | 3 | 61 |
| | Female | 4 | 13 | 10 | 0 | 27 |
| Total | | 19 | 39 | 27 | 3 | 88 |
| | $\chi^2$ (3) = 2.79, ns | | | | | |
| | $\chi^2$ (2) = 1.4, ns | | | | | |

*Note*: Eight participants did not answer this question.

### Psychological measures

In the current study, the internal reliability of the HSCL-25 Anxiety Subscale was Cronbach's $\alpha$ = .89. For the HSCL-25 Depression Subscale, Cronbach's $\alpha$ = .80 and the PCL-5 scale had Cronbach's $\alpha$ = .89. The correlation between the two HSCL-25 subscales was .55 ($p < .001$) and the correlation between the full scale HSCL-25 and the PCL was .62 ($p < .001$). Correcting for the restriction in range in the sample, using Thorndike's (1949) method, the PCL-5 correlation with the HSCL-25 was .95, indicating that the two measures were tapping into the same construct.

Before turning to the main analysis, it needed to be established that the random assignment of participants to groups had been successful. A one-way analysis of variance (ANOVA) of the HSCL-25 scores by treatment group at Time 1 was not significant, $F(2, 287)$ = .65, ns. This indicates that the random assignment process was successful.

### Main analyses

The central question asked by the study was whether the treatments (HROC or PVE) significantly reduced anxiety and depression (as measured by the HSCL-25) in the participants compared to the waitlist control group. A mixed-design ANOVA was conducted to answer this question. There were four assessments, three groups of participants and two sexes. For this analysis, the total number of subjects with complete data was 253, and the dependent variable was the HSCL-25 mean scores.

The principal analysis produced three significant main effects: sex, experimental condition and assessment and one two-way interaction. The main effect for sex ($F = 1, 247$) = 4.58, $p < .04$) involved males having higher overall HSCL-25 scores ($M = 2.19$) compared to females ($M = 2.07$). The main effects of the experimental condition and assessment were also significant but are best understood in the context of the significant interaction of treatment group × assessment ($F = 5.88$, df = 5.88, 725.51 [using Greenhouse–Geiser correction], $p < .001$). Figure 1 shows the graph of the interaction.

Additional analyses found that this pattern was true for both the anxiety and depression subscales on the HSCL-25.

The HROC/PVE and PVE/HROC groups were analyzed independently of the control group to better understand the interaction. Recall that the PVE group was considered an active control group. An analysis of covariance (with sex as the covariate) found that these two conditions (i.e., the main effect) were not significantly different from each other overall, $F(1, 170)$ = 0.01, ns. In addition, the interaction between the two conditions and the assessment factor was not significant, $F(2.97, 504.13)$ = 3.54, ns. Therefore, the two experimental conditions were combined for the next set of analyses.

Comparing the combined treatment groups with the control group over the four assessments and using the HSCL-25 mean scores as the dependent variable and sex as a covariate produced a significant interaction, $F(2.94, 734.88$, using Greenhouse–Geisser correction) = 7.46, $p < .0005$. See Figure 2 below. The combined treatment groups reduced symptoms more quickly and to a greater degree over time than the waitlist control group.

To better understand the nature of the interaction, a series of interaction contrasts were performed, enabling an understanding of the impact of the combined HROC + PVE and the control group between Time 1 and Time 2 (after the first workshop), Time 2 and Time 3 (after the second workshop) and Time 3 to follow-up Time 4. These were 2 × 2 comparisons, with one factor being the two intervention groups (i.e., the combined treatment groups and the control group), and the second was the two assessment times. The first of these analyses produced a significant interaction, $F(1, 296)$ = 11, $p < .001$. Given that the two treatment conditions were not different at the pretest Time 1 ($t(296)$ = 0.91, ns), the

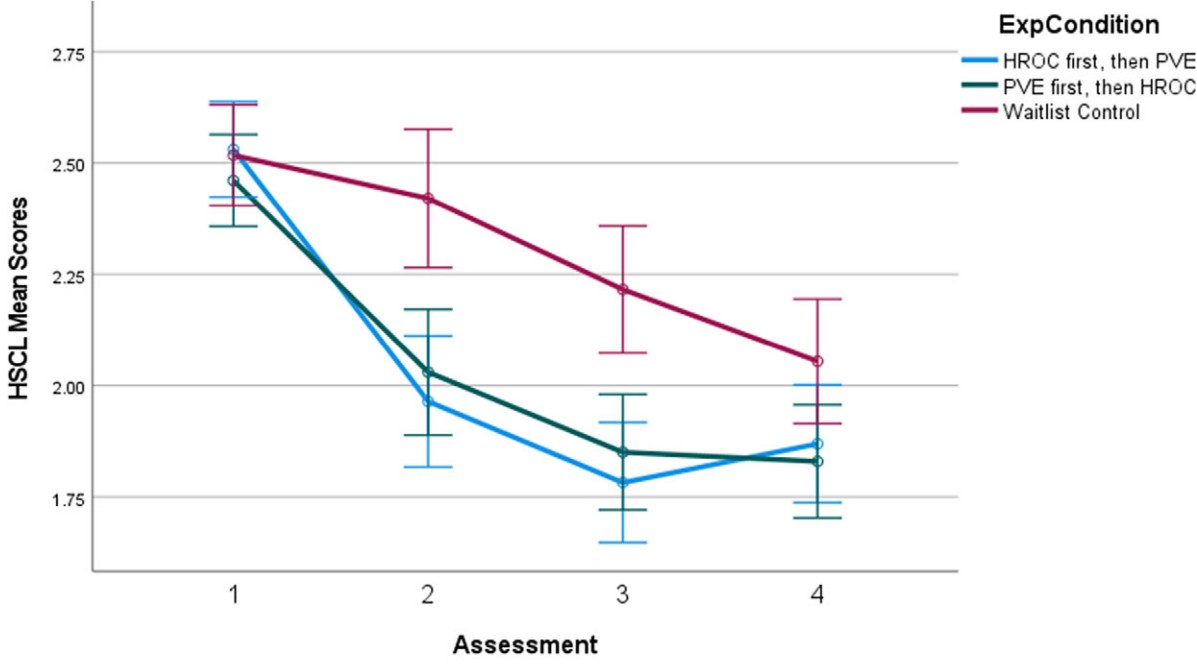

**Figure 1.** Results for three experimental groups by four assessments on mean HSCL-25 scores.

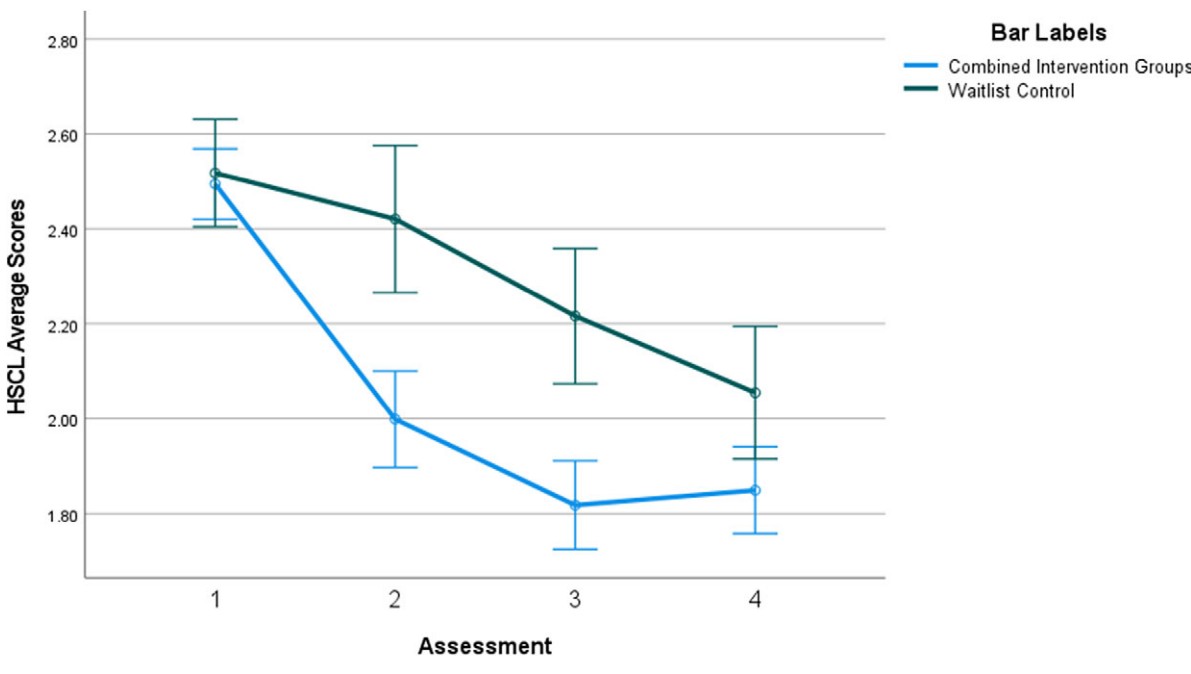

**Figure 2.** Combined workshop groups compared to the waitlist control group on mean HSCL-25 scores.

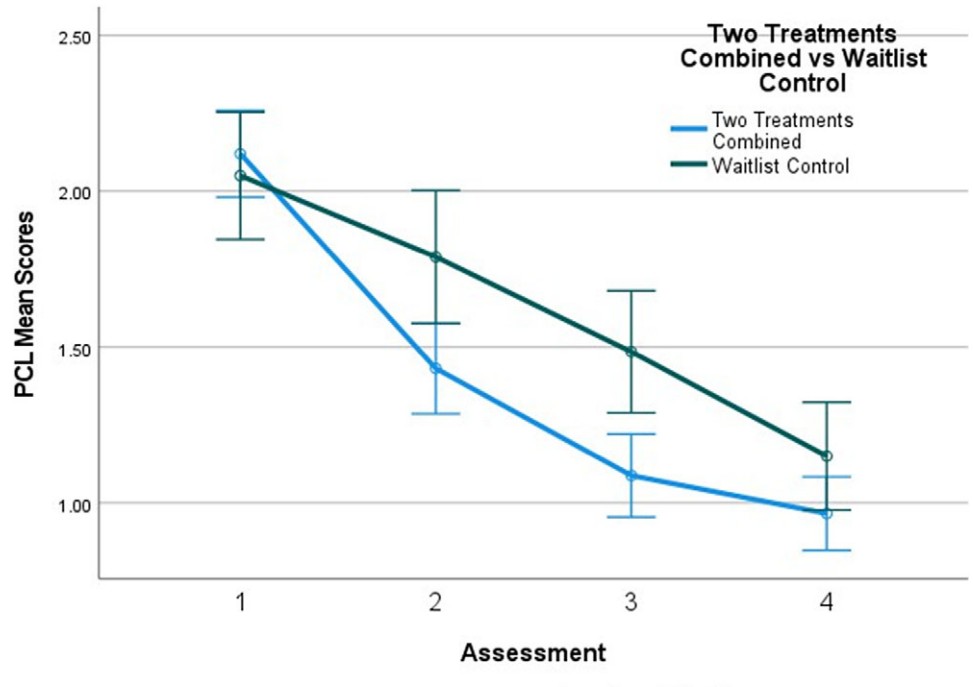

**Figure 3.** PCL-5 mean scores for combined treatment groups and waitlist control group.

interaction indicated that the groups did differ at the second assessment ($t$ (296) = 4.14, $p$ < .001). The combined treatment group had significantly greater symptom reduction than the control group. Figure 3 illustrates that the combined treatment group had significantly lower symptom severity scores on the HSCL-25 than the control group at Time 2.

Moving to the second interaction contrast, the main effect of the combined intervention group versus the waitlist control group was significant, $F(1, 294) = 21.80$, $p$ < .001, but the interaction was not $F$ (1,294 = 1.03, $p$ > .05). This result indicated that the difference between the combined treatment group and the control group comparison found in Assessment 2 was maintained in Assessment 3.

The final interaction contrast produced a significant interaction, $F(1, 251) = 7.77$, $p = .006$. At the third assessment, the groups were significantly different, $t (294) = 5.66$, $p < .001$, with the combined treatment group having significantly lower anxiety/depression severity scores than the control group. This group difference was still significant at the fourth assessment, $t (253) = 2.45$, $p = .02$, but to a lesser extent than at the third assessment. Thus, symptom reduction was maintained, but to a lesser degree.

To summarize, the level of symptom distress at baseline in the two combined treatment groups (HROC and PVE) and the control group started equal to each other. Both HROC and PVE proved effective in reducing anxiety/depression compared to the control group. These improvements were maintained after the second treatment was delivered. By the 3-month follow-up, the differences were still significant but less so.

Finally, the impact of the workshop interventions was examined on the PCL-5 scores. The interaction between the combined treatment groups and the control group was significant, $F(5.78, 719.58,$ with Greenhouse–Geisser correction) $= 3.70$, $p = .009$. The results of this analysis paralleled those obtained when examining the HSCL-25. They are displayed in Figure 3.

## Discussion

The study has demonstrated that, even in conflict zones, important steps can be taken to reduce the suffering of the local population. The HROC intervention significantly reduced anxiety, depression and traumatic stress symptoms for up to 4 months compared to the waitlist control group. There is substantial support in the global mental health community for the efficacy of psychosocial and brief psychological interventions that can be delivered by trained laypeople (Murray et al., 2011; WHO, 2015; 2022; 2023; Haroz et al., 2020; Okoroafor and Christmals, 2023). The current study is an example of how global mental health strategies, particularly research, evidence-based care, cultural adaptation and task shifting, may be implemented to broadly address traumatic distress and mental health concerns. This study employs a systematic empirical approach to evaluate the efficacy of psychosocial and peacebuilding interventions in the context of limited resources. The anxiety, depression and trauma symptom measures used in this study are consistent with other studies utilizing them in other international contexts (Lima et al., 2016; Mughal et al., 2020). The measures used in this study were also supported by the Patel et al. (2020) study from the same community examining anxiety, depression and traumatic stress symptoms.

Initial support for the HROC intervention came from Yeomans et al. (2010) and unpublished, informal feedback from earlier international cohorts. Participants reported reduced symptomatology, gained a better understanding of their symptoms and requested additional workshops to continue their recovery. What was more surprising was the impact the PVE workshops had on mental health variables. The PVE workshop was designed to inoculate participants against genocidal ideology and to make clear that there were stages to increasing aggression to genocidal levels and barriers to escalation through engaging in different positive dialogues and behaviors for peaceful solutions.

The significant differences between the HROC intervention group and the waitlist control provide empirical support for this intervention. However, the PVE intervention group provides crucial information on how we interpret the impact of the HROC intervention. The results indicated that the two treatment groups did not significantly differ. As noted earlier, the PVE was intended to be an active control group. Its subject matter focused on the CAR's colonial history and episodes of repeated mass violence that could lead to genocide. It aimed to discuss ways of identifying and intervening in the cycles of violence that lead to genocidal ideology. When typically deployed, PVE target outcome questions were related to changes in social relationships, attitudes toward conflict resolution and revenge versus dialogue. In contrast, the HROC intervention focused on education about distress symptoms as an expected reaction to violence, grief, loss and anger, as well as the commonality of experience in civil war and how the symptoms could be managed. Thus, the interventions significantly differ in their content. Examples of this may be found in the comments of workshop participants, which clearly distinguished which workshop a participant attended.

While the substantive content of the two interventions focused on different topics (i.e., individual psychological distress vs. resisting or accepting retributive social norms), the two interventions produced comparable and positive outcomes (compared to the waitlist control). Explanations are likely to lie in the common group factors employed in the two interventions (McAleavey and Castonguay, 2014; Patel et al., 2020). Common factors found in treatment manuals may explain why our distinct interventions yielded similar results. Common factors research in psychotherapy outcomes has been discussed for decades, and more recently has been applied to brief psychosocial interventions in global mental health research. It may help explain why so many different psychosocial interventions in LMIC produce positive results.

LMIC psychosocial interventions are dependent upon the existing infrastructure and personnel. If infrastructure/personnel are minimal, then laypeople trained to deliver services through task shifting are utilized. With greater infrastructure/personnel, task sharing may be undertaken with mental health supervision of trained laypeople. These different models of service delivery often dictate the length of treatment that can be accomplished, usually between 1 and 18 sessions, but averaging around 8 sessions. In addition to this, earlier research on psychosocial interventions began with limited sessions or workshops. More recently, treatments have expanded to include transdiagnostic, symptom cluster-specific problems that can be treated in the short-term as needs arise (Murray et al., 2014; Singla et al., 2017). Furthermore, recent research (Patel et al. 2016; Singla, 2021) suggests that the evidence gathered from global mental health work has demonstrated that psychosocial and targeted symptom-cluster interventions can be utilized in the same way in High income countries (HIC) that also have a limited supply of professionals for great demand and cost.

Our interventions fall into the former category of workshop interventions in a post-conflict/conflict environment. There was essentially no infrastructure, and one general medical provider with some mental health training. Training trusted community members was a necessity. The context also meant that providers and participants shared quite recent and similar experiences. Participants were very motivated to receive help, and likely their expectations were heightened by the urgency of their need. The match between provider-community members and participant-community members is akin to therapist/client matching, which provides a preordained congruence in the relationship. This supports the common factor of relationship and expectations of help. Wampold (2015) explores how empathic communication is central to establishing the relationship.

Cuipers et al. (2019) take exception to the conclusion that common factors are involved in why different interventions have comparable outcomes, pointing out that not all studies demonstrate this finding. To resolve these differences, multiple, very large (for power) and complicated experimental studies must be carried out to systematically look at different process variables involved in outcomes.

Regardless, Cuipers is identifying meta-analyses that involved what are called "bona fide" treatments of greater duration than our interventions. In our case, 2- to 3-day workshops may be more likely to involve the common elements because of the context (relief from ongoing violence) and the level of need for relief from psychological distress. As such, there may be great hope and optimism for being attended to, being acknowledged for their suffering and sharing with others in their community so that the commonality of experience is relieving.

Yeomans et al. (2010) provide further evidence of process components in HROC. Their study administered HROC under two conditions against a waitlist control group. Groups attended HROC that included psychoeducation about post-traumatic stress and another version of HROC that did not include psychoeducation about trauma. To make both HROC interventions equivalent in time, investigators added exercises devoted to social engagement. The exercises included pairing participants and having them answer or discuss questions provided to them. The exercise facilitated greater communication about trust, safety, sense of security and interethnic conflict. Discussion questions, such as "someone I trust and why" and "a time I overcame fear," were used and the pairs were asked to share information about how interethnic conflict had impacted each of them. While both HROC groups experienced a reduction in symptoms as measured by the HSCL-25, the group that improved the most was the group with enhanced social engagement. The increased time devoted to enhancing the interpersonal relationship through dialogue benefited and reduced subjects' distressing symptoms more than HROC with PTSD psychoeducation.

Whether the benefits observed were due to common factors or due to specific aspects of each intervention is a question that warrants additional evaluation and may guide future research. That said, the value of these interventions is clear, as illustrated by the comments of one study participant: "*I called him 'my brother'… He was surprised! Then I continued: 'It would have been your end today if I had not been in a workshop and learned about the consequences of violence and steps leading to it. Come with me tomorrow morning and attend the last day of the workshop: you will understand that you were wrong being in a militia killing people.'*" There were also requests from militia leaders for their children to experience PVE treatment, even though they themselves would not. They felt they were caught up in the hostilities but hoped for a better life for their children.

Returning to the combined workshop groups compared to the waitlist controls, the two intervention groups yielded similar results. Therefore, they were combined for analysis and compared to the waitlist control group. These two groups – intervention and control – started with a similar symptomology (depression, anxiety and trauma), but the intervention group experienced a significant decline in symptoms compared to the control group. The control group also reported a gradual decline in symptoms, but the intervention group experienced significantly fewer symptoms than the control group. The gradual improvement of the control group was not anticipated but might be due to several factors. We consider two possible explanations for this finding: first, if there is a reduction in conflict, then lives stabilize and emotional distress returns to baseline levels. At the time of this study, there were UN peacekeeping troops on the ground that served to reduce the level of conflict. At the same time, the research team heard gunfire twice during the study, and they were not allowed to go outside of Bangui because it was judged too dangerous. This instability is reflected in an overview of the conflict (International Peace Information Service, 2018). Another possible explanation for these outcomes is that the repeated interviews in and of themselves acted as an intervention since individuals on the waitlist were given the opportunity to share their experiences. Similar waitlist control effects were observed in Patterson et al. (2016). In future studies, it may be prudent to add additional waitlist control groups with fewer interviews.

Since this study was conducted, more advancements have occurred in the Global Mental Health field. Systematized training and monitoring paradigms have been developed to train community-based or nonspecialist health workers. The WHO has developed manualized interventional training on various mental health topics (WHO, n.d.). The Ensuring Quality in Psychological Support is a joint WHO and UNICEF online training tool that includes all e-learning courses and competency assessment tools for scaling up community health workers in the key areas of depression and anxiety (Kohrt et al., 2024). In addition, a more sophisticated strategy for demonstrating training and adherence to common elements practices in psychosocial healing has been demonstrated with the enhancing assessment of common therapeutic factors (Kohrt et al., 2015). Similarly, the Common Element Treatment Approach (CETA) methodology developed by Murray et al. (2014) requires supervision by mental health professionals of trained laypeople. However, these more recent advancements presuppose infrastructure, personnel and trainers that do not exist in CAR.

It should be noted that a distinction must be made between different dimensions in the implementation of interventions. That is, is it early/later in the cessation of conflict/disaster, what is the infrastructure and what personnel are available and is the intervention community-based or individually based? The Interagency Standing Committee on Mental Health and Psychosocial Support identifies a continuum of service delivery (IASC, 2006). This parallels the development of interventions across time in the literature. Generally speaking, post-conflict results in degradation of infrastructure. Preexisting conditions in the country relate to what personnel may be mobilized. In the CAR, there was a paucity of infrastructure as well as almost nonexistent personnel. As such, immediate community-based interventions were carried out. Our study focuses on brief psychosocial interventions early in recovery. Targeting regionally based training could enhance service provision so that both sustainability and interventions could move beyond the immediate capital area.

A limitation to the generalizability and replicability of this study in other settings is that community partnerships were critical to our ability to bring together formerly warring groups, who were essential to this study. Without their cooperation and support, the study could not have happened. In addition, this study may only be sustainable with funding from NGOs and other donors. Although the study is of low cost and involves laypeople, significant sustainability issues exist. Providing the training is dependent on funding and psychologically minded NGOs. In-country trainers may come from around the country, so transportation and lodging costs must be covered. Meeting rooms must be rented, travel costs for participants must be reimbursed and food and drink need to be provided. Ongoing training and support for the trainers are essential as well, as this is taxing work.

Sustainability may also be addressed by looking to literature on digital interventions using computer technology. For example, both individual and group interventions have been shown to improve depression, maternal/infant health, smoking cessation and other mental health disorders (Muñoz et al., 2018). Unfortunately, access to telecommunications in CAR is both expensive and the infrastructure is lacking (International Telecommunication Union, 2023). When it does become practically available, one of the important questions to ask will be whether advances in communication impact the outcomes found in this study.

Another limitation is related to the feasibility of such a study in the context of ongoing conflict. Beyond the preexisting challenges to mental health care in under-resourced communities, ongoing conflict presents a host of additional challenges to trauma healing. Studies on the efficacy of trauma treatment have found that interventions are less effective when those in treatment continue to be exposed to traumatic stress (Bass et al., 2013). Moreover, when resources are scarce, it is often deemed unreasonable to pursue interventions aimed at trauma. A community in the throes of a conflict has more pressing needs related to the physical safety of the public, and this applies to the interventionists as well (Bolton et al., 2007). Nonetheless, group psychotherapy for the treatment of traumatic stress symptoms is still effective in communities that continue to experience traumatic events (Bass et al., 2013; Resick and Monson, 2024). The issue of ongoing conflict is certainly relevant to CAR. At the time of this study, the conflict had mostly stopped, and it seemed as though CAR was going into a relatively peaceful time. This turned out not to be sustained peace. However, it raises the question of whether this work could have been done amid conflict and whether warring factions could have been brought together in this way had there not been a sense of hope that they were entering a period of peace.

## Conclusions

The findings from this study provide support for the practice of training and supporting local paraprofessionals in the implementation of short-term, culturally appropriate, evidence-based interventions. These findings suggest that these interventions can reduce symptoms related to anxiety, depression and traumatic stress symptoms in post-conflict settings. However, the longevity of these effects may depend on how stable the environment is and what other activities the participants engage in. It is imperative to continue assessing the impact of psychologically related interventions. Funding agencies understandably prioritize the emergency needs of the populations in crisis. Mental health needs are typically not addressed first, although they are crucial to getting the local economies functioning again and improving the lives of the affected citizens. Given that resources are often limited in post-conflict settings, it is important to maximize the usefulness of any mental health intervention. Understanding which mental health interventions work and which do not is vital. Further, even interventions (e.g., PVE), which were not expected to have an impact on mental health, turned out to be effective, which raises the very important question as to what the active elements are in the mental health interventions that work. The fact that some effective interventions can be delivered by paraprofessionals at low cost means that the help that once seemed unattainable is within reach.

**Open peer review.** To view the open peer review materials for this article, please visit http://doi.org/10.1017/gmh.2025.10015.

**Data availability statement.** Due to the sensitive nature of participant responses, data are not made available to the general public. Researchers can request the sharing of data from W.J.F.

**Acknowledgments.** The authors specially thank all members of the Catholic Relief Services staff, in particular Robert Groelsema, Driss Moumane, Ariana Proctor, Florence Ntakarutimana and Olive Somse; Aegis staffs Freddy Mutanguha, Alain Lazaret, Lambert Kanamugire and Glen Ford; Echelle Translation Services staff Alain Serge Magbe and Dr. Lisa Brown of Palo Alto University. Natasha Greenberg of USAID believed in the project and helped get it funded. Dr. Chad Murchison provided expert statistical consultation.

**Author contribution.** Conceptualization: W.J.F., K.B.F., S.G.P.; Data curation: W.J.F.; Formal analysis: W.J.F.; Funding acquisition: K.B.F., Glen Ford (Aegis Trust), W.J.F.; Investigation: W.J.F.; Methodology: W.J.F.; Project administration: W.J.F., K.B.F.; Supervision: W.J.F., K.B.F.; Writing – original draft: W.J.F., M.V.Z., K.B.F., V.B., S.G.P.; Writing – review and editing: W.J.F., M.V.Z., K.B.F., V.B., S.G.P.

**Financial support.** The research project was funded directly by a contract to Palo Alto University from the Catholic Relief Services as part of a grant from USAID (USAID RFA-OAA-15-000017). A combination of funders provided necessary direct and indirect support and donations in kind. These organizations were USAID, Aegis Trust, Global Inter-Religious Alliance, Plateforme Des Confessions Religieuses de Centrafrique, Islamic Relief and World Vision. The Sea Grape Foundation provided important matching funds in the public/private funding of this project.

**Competing interests.** The authors declare no competing interests exist.

**Ethics statement.** This study was approved through the IRB at Palo Alto University. Informed consent was obtained from all participants, including information explaining that data would be anonymized to preserve the privacy of participants. Information was provided as to the rationale for the study and storage of data.

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
