## [Reviewer Report]

The article makes some important points and presents useful analyses in terms of what works in humanitarian and crisis settings. Its also points out very important arguments regarding sustainability, significance of contextually informed interventions, and dependence on outside NGOs. Some suggestions for improving the manuscript:

Overall, the flow of the article is a bit hard to follow. some of the sections seem to be overlapping, or mismatched (eg results in methods, methods in results), and its hard to understand how the authors get to their conclusions in the discussion given the lack of attention to common factors and process elements in the intro and methods. Some other specific feedback:

1. clean up the aims to clarify what you are testing. Specifically, the wording for the sentence starting on line 190, “if both HROC and PVE produce similar positive results, then the common elements of the two…” —common elements have not been introduced, and this is an assumption. I would be specific in defining what common elements (or factors) are. Additionally, it’s not clear how or whether the group vs individual focus of the interventions is assessed as a component for symptom change, and additionally, it’s not clear about what skill sets of the workshop facilitators have that could influence outcomes, and whether those are assessed or not.

2. add a table to show the differences in components for the two interventions, and consider describing the two interventions in the methods section.

3. methods - interviews are introduced but not explained, are they one-on-one, or qualitative? the assumption is that they are one-on-one to fill out surveys, this should be clarified.

--can you say more about the validity of the tools in general, as well as the impact of not having the tools translated but rather read out in one language or the other? how was the translation confirmed for the items, what process was used?

—please describe in more detail the consent and recruitment process, as well as the randomization (eg blinding or no blinding, who was it done by).

4. results - the first 3 sentences are methods and should be moved there. same for the exploratory methods on gender.

5. discussion — it’s unclear why the authors would jump to CETA besides the trauma module, there isn’t indication from participants that they would want or need further individualized TX, particularly as this study couldn’t differentiate between the group and individual-focused TX outcomes.

- the authors suggest that there are process elements and common factors that are the cause for the outcomes, which could be true, but the authors did not seem to present, code, or analyze distinct common factors or process elements. This would be a key next step (eg moderation or mediation analyses). Additionally, regarding contextual and cultural reflections as the authors mention a few times, it’d be a critical next step to interview qualitatively with the participants to determine what they find as the most impactful elements, or conduct something like a theory of change workshop, and again, whether more individualized focused trauma care is useful.

It’s confusing why so much description is given to CETA in the discussion, given the amount of time the authors highlight, in honesty, the lack of resources to sustain such a rigorously implemented form of care and highlight how impactful the two psychoeducation-based workshops are. It could be useful for the authors to consider other routes or options in the discussion, as well as to consider future steps in terms of answering what could work next, including further analyses on common factors, qualitative and stakeholder interviews, and potentially considering transitional justice reform or larger policy changes that are needed, including who and how these interventions are used. For instance, it would be great to hear the authors' thoughts on whether there are ways to sustain this delivery outside of psychologically-minded NGOs dropping in and leaving again, for instance via train-the-trainer models and reimagining traditional or colonialist-informed clinical supervision models, and whether trauma care is something different in this setting than the western clinical models currently dominating the world.

---

## [Reviewer Report]

This is an interesting study and a contribution to the literature, particularly with regards to interventions in CAR. There are a number of areas where more detail is needed and the analytic plan needs to be revisited since the intervention was group based rather than individual. In addition, the finding that effects were not maintained over time requires more discussion. The results seem overstated, as the intervention was not more effective than waitlist control at follow-up.

Page 7, line 112: The authors note that the measures are “cross-culturally validated.” Please provide details about the validity testing of the HSCL, HTQ and PCL in CAR.

Page 8, line 135: The authors note that group composition should consist of both Christians and Muslims (both sides of the conflict). Did this occur? Please provide a demographics table of the whole sample and then for each of the 3 conditions. How many members of each of the groups were in each of the workshops?

Page 11 Participants:

- Please provide more details about who the religious leaders were and how and why they were selected to nominate individuals. Were the data derived from just a few religious communities? How many people were self-nominated and how did they hear about the study?

- Please provide a flow chart explaining the reduction in participants from 960 to 290.

- Were there any differences between the 260 who completed the program and those who did not?

- There appear to be more Christians (67%) than Muslims (27%). What are the implications for the intervention? What was the background of the facilitators?

Page 12 line 216: The authors note that transportation costs were paid to those who participated at the office or community space. Were any incentives provided for the assessments themselves? For the workshops?

Page 12, line 221: What was the translation process and who did the translations? How many participants completed the interview in Sango vs. French?

Page 12, Measures and Inclusion Criteria: Was the HSCL cut-off the only inclusion criteria? Why was the HSCL used and not the PCL if the primary outcome was PTSD symptoms? Were there any exclusion criteria? How was the cut-off score determined? Line 234: “a subset was scheduled and could participate as required.” Why just a subset? How was this subset selected?

Page 13 line 238: Please provide the N sizes and not just the percents. A table of trauma events would be helpful.

Page 13, line 248: The authors state “the proposed study was discussed with a psychology professor at the University of Bangui.” What does this mean? What happened in the discussion? what was the purpose and outcome of the discussion? Was the professor involved?

Page 13 line 250: In what ways was the Consortium of Interfaith Peacebuilding Platform involved in the study?

Study Design:

- Who led the groups/workshops? what was their training and supervision? How were they selected?

- Provide more detail about the workshops. Where were they conducted? How long were they? Were participants compensated for participating in or traveling to the workshops?

- Who were the assessors? were they blind to participant condition?

- How was randomization done?

- How many workshops were conducted and how many people were in each group?

- This was a group-based treatment and so analyses should be conducted at the group level rather than the individual level. Multi-level models?

- Were the same participants together in each group, i.e., if they completed HROC first were they with the same people to complete the PVE?

Results

- Line 264: Was the final sample 290 or 253? Was the analysis only based on the complete cases? Why not use an intent-to-treat approach? Were there differences between those who dropped out and those who completed?

- Line 268: “the subject’s sex was not involved in any significant interactions” What does this mean? If sex predicted the primary outcome then it makes sense to include it as a covariate

Discussion

- Line 371: “Examples of this may be found in the comments of workshop participants, which clearly distinguished which workshop a participant attended.” What comments of workshop participants? to the assessors? If so, then were the assessors actually blind to condition?

- Line 427: who were the “key stakeholders and community partners” and what was their involvement?

- Line 431: Who were the in-country trainers and what was their involvement?

- More needs to be said about the fact that differences were not observed between the waitlist condition and the intervention condition at 3 months. In what ways does this reduce enthusiasm for the interventions or not?

Figure 1

- In the legend, list the interventions as the combined HROC/PVE and PVE/HROC

- add 95% CI to the graph

---

## [Reviewer Report]

Thanks to the authors for the revisions. This work remains imporant and it is clear that the authors implementing the interventions are quite knowledged about the processes and their impact.

The authors’ edits have improved the article as a whole, including adding more tables and descriptions to support the overall goals of the article. However, a few comments go unaddressed and the article could benefit from further revision, particularly to clarify the approach in formulating the hypothesis and goals of the research, and connecting to recent, relevant global work.

A few specific comments:

1.The authors must elaborate on the common elements argument via existing evidence, ideally both in the intro or the discussion. For instance, the common factors has empirical evidence and huge impact to the field over the last 10 years, especially the last 6 (e.g., ENACT tool, “EQUIP”, reviews by Singla et al, Wampold et al, Cuijpers et al, and other researchers in the common factors field). It’s unclear why the authors do not delve into this argument, as was suggested in one of the reviewers’ comments, as it would support the research study more strongly. I suggest the authors consider strengthening their argument about common factors as effective components by citing relevant literature from the field to bolster their hypothesis and goal of the research itself.

2. similarly, the argument that the two interventions are uniquely different via its trauma-focus remains slightly undermined by how the authors compare and describe the two. under the present study, the authors outline that PVE isn’t intended to improve MH outcomes thus allowing it to act as an “active control.“ However, going through the article, it seems perhaps the difference is simply in how mental health or trauma is framed when the intervention was developed, leaving questions as to whether PVE is doing trauma-informed work but without the Westernized lens? For instance, when authors describe PVE, they lay out that it uses mental health techniques, ”cognitive and behavioral skills “, and the outcome “to engage in positive activities in the future”. The comparison table shows that PVE uses common factors, or training in mental health foundational helping skills, such as empathy & active listening . Additionally, in the section ”comparing interventions", HVOC is described as different because it discusses history and concept of collective trauma for the individual versus PVE focusing on a genocide (a collective traumatic event) and discussing shared histories of cycles of violence such as colonialism. How are these components uniquely different? Both are group-based. And both seem to offer coping skills based on the authors’ descriptions. As such, the authors could clarify what they mean by not being different since one is not “trauma-focused”. For instance, describing what it means to for interventions or psychosocial support to be trauma-focused, and how one would know it wasn’t, especially given the group-based context.

3. the table comparing interventions needs clearer column and row headings

4. suggest to add numeric psychometric properties such as internal consistency, construct validity, and validated cut-offs for the measures.

5. similar to the intro, the discussion writes about common factors but does not include any recent (the last 1-3 yrs) citations. Given common factors is evidenced so widely in the field, the article could be improved by linking to empirical research in more depth, such as providing explicit examples of how it’s been identified as effective when compared to tx specific elements, and how much attention has been given in the last 6 years alone on concretely training non-specialists and specialist alike, (particularly in humanitarian settings), in these skills and group-facilitation skills.

6. The authors do not aptly elaborate in the discussion as to why a more intensive, individualized, resource-heavy trauma intervention, like CETA, is the appropriate next step whilst they accurately describe the context as "significant sustainability issues exist. Providing the training is dependent on psychologically minded NGOs. In-country trainers may come from around the country, so transportation and lodging costs must be covered. Meeting rooms must be rented and food and drink need to be provided. Ongoing training and support is essential as this is taxing work.

Another limitation is related to the feasibility of such a study in the context of ongoing conflict. Beyond the pre-existing challenges to mental health care in under-resourced communities, ongoing conflict presents a host of additional challenges to trauma healing.”

The nod to group psychotherapy is useful, but also lends the question as to why a group-based psychotherapy intervention wouldn’t be the logical next step vs an individual-based trauma therapy like CETA.

7. Overall, the original supposition of the article, that common elements/factors are important and, trauma-focused or not, the intervention helps folks because of common elements/factors, remains unclear since it was not analyzed specifically for this study. It would be ideal to discuss other critical next steps, such as more in-depth quant or qual analysis into the impactful and effective components of each intervention.

---

## [Reviewer Report]

I thank the authors for their attention to addressing the reviewer comments. There are still some areas that I recommend be improved prior to publication:

1. The details of the sample are a bit conflicting:

- One abstract says 298 participants, the other says 290.

- In the methods it says “The initial pool of participants was 960 adults who lived in Bangui” and then later it says “The initial testing involved 1103 individuals. The first set of participants (N = 143) were used as pilot subjects for the assessors in order to clear up any questions about testing procedures and meaning of the questions. The formal testing was then conducted on 960 participants.” I suggest starting with the 1103 and then explaining how you reduce down to 960. Why was the sample size of 143 selected?

- Why were only 450 participants called out of the 502 who met the HSCL cutoff of 1.75? What happened to the other 52 people?

- Sometimes the authors state that 650 people met the HSCL but off, and other times it says that 502 met the cut-off. Which is correct? Please provide a participant flowchart starting with the 1103 and ending with the 290 indicating how many participants dropped out or were removed from the study at each step.

2. The mechanisms underlying the two interventions are still not well-fleshed out. The HROC intervention is defined as having a “trauma-focus” but the outcome is anxiety and depression symptoms. Please describe the mechanisms of the HROC intervention that are theorized to be the “active ingredients” of anxiety and depression symptoms and why these factors are not theorized to be present in the PVE intervention.

3. At a few points the HSCL is referred to as HSRC. Please correct and spell out HSCL the first time it used.

4. The authors state “The final number of participants was 290, defined as people who participated in at least 3 of the 4 assessment sessions.” How many participants participated in each workshop and each assessment session?

5. Were there any differences in the key variables and demographics of those who were initially screened, those who met inclusion criteria but did not complete the sessions, and those who completed the full study and were included in the final sample of 290 participants?

6. Why were participants not scheduled for the next assessment if they missed the preceding workshop? This process does not align with best practices of an intent-to-treat approach.

7. The decision to use and anxiety/depression measure (the HSCL-25) as the screening criteria for a PTSD intervention is still unclear. There is no literature cited to support the statement “however the HSCL-25 has been used more internationally and is easier to understand compared to the PCL-5.” In addition, the Barbano et al 2019 citation was not listed in the reference list so I could not check the cited reference that the HSCL “targets two major inherent components of PTSD.”

8. The justification for sample size of “we wanted to test as many subjects as possible” is not adequate. Please provide more information to justify the sample size.

9. The authors note “This study was part of a larger effort to de-escalate the violence gripping CAR. The CIPP was an interfaith group composed of Catholic Relief Services (CRS), Aegis Trust (AT), Islamic Relief Worldwide (IRW), and World Vision International (WVI). The function of the group included three major foci: 1. Civil Institutions Establish Leadership Role in Peacebuilding; 2. Livelihoods Security is Re-established; 3. Social Cohesion Fostered through HROC and PVE.” It is not clear what CIPP is or how these organizations interfaced with the study.

10. There are repeated and redundant sections throughout the manuscript.

---

## [Editor Report]

Dear Dr. Froming,

Thank you for submitting the revised manuscript entitled “RCT of Post-Conflict Trauma Interventions in the Central African Republic” to Cambridge Prisms: Global Mental Health. We appreciate the effort you have put into addressing the comments from the previous round of reviews. After a careful re-evaluation of the revised manuscript, we would like to offer the following additional feedback, which we hope will help guide you toward further enhancing the clarity and quality of your article.

The reviewers have provided detailed and constructive suggestions that we strongly encourage you to carefully consider. We recommend that you respond thoroughly to each of the reviewers' comments, providing justifications for any suggestions that you choose not to incorporate. If you do decide not to implement specific feedback, please explain why this decision was made, ensuring that your response reflects a thoughtful and evidence-based approach.

Reviewer 1:

Thanks to the authors for the revisions. This work remains imporant and it is clear that the authors implementing the interventions are quite knowledged about the processes and their impact.

The authors’ edits have improved the article as a whole, including adding more tables and descriptions to support the overall goals of the article. However, a few comments go unaddressed and the article could benefit from further revision, particularly to clarify the approach in formulating the hypothesis and goals of the research, and connecting to recent, relevant global work.

A few specific comments:

1.The authors must elaborate on the common elements argument via existing evidence, ideally both in the intro or the discussion. For instance, the common factors has empirical evidence and huge impact to the field over the last 10 years, especially the last 6 (e.g., ENACT tool, “EQUIP”, reviews by Singla et al, Wampold et al, Cuijpers et al, and other researchers in the common factors field). It’s unclear why the authors do not delve into this argument, as was suggested in one of the reviewers’ comments, as it would support the research study more strongly. I suggest the authors consider strengthening their argument about common factors as effective components by citing relevant literature from the field to bolster their hypothesis and goal of the research itself.

2. similarly, the argument that the two interventions are uniquely different via its trauma-focus remains slightly undermined by how the authors compare and describe the two. under the present study, the authors outline that PVE isn’t intended to improve MH outcomes thus allowing it to act as an “active control.“ However, going through the article, it seems perhaps the difference is simply in how mental health or trauma is framed when the intervention was developed, leaving questions as to whether PVE is doing trauma-informed work but without the Westernized lens? For instance, when authors describe PVE, they lay out that it uses mental health techniques, ”cognitive and behavioral skills “, and the outcome “to engage in positive activities in the future”. The comparison table shows that PVE uses common factors, or training in mental health foundational helping skills, such as empathy & active listening . Additionally, in the section ”comparing interventions", HVOC is described as different because it discusses history and concept of collective trauma for the individual versus PVE focusing on a genocide (a collective traumatic event) and discussing shared histories of cycles of violence such as colonialism. How are these components uniquely different? Both are group-based. And both seem to offer coping skills based on the authors’ descriptions. As such, the authors could clarify what they mean by not being different since one is not “trauma-focused”. For instance, describing what it means to for interventions or psychosocial support to be trauma-focused, and how one would know it wasn’t, especially given the group-based context.

3. the table comparing interventions needs clearer column and row headings

4. suggest to add numeric psychometric properties such as internal consistency, construct validity, and validated cut-offs for the measures.

5. similar to the intro, the discussion writes about common factors but does not include any recent (the last 1-3 yrs) citations. Given common factors is evidenced so widely in the field, the article could be improved by linking to empirical research in more depth, such as providing explicit examples of how it’s been identified as effective when compared to tx specific elements, and how much attention has been given in the last 6 years alone on concretely training non-specialists and specialist alike, (particularly in humanitarian settings), in these skills and group-facilitation skills.

6. The authors do not aptly elaborate in the discussion as to why a more intensive, individualized, resource-heavy trauma intervention, like CETA, is the appropriate next step whilst they accurately describe the context as "significant sustainability issues exist. Providing the training is dependent on psychologically minded NGOs. In-country trainers may come from around the country, so transportation and lodging costs must be covered. Meeting rooms must be rented and food and drink need to be provided. Ongoing training and support is essential as this is taxing work.

Another limitation is related to the feasibility of such a study in the context of ongoing conflict. Beyond the pre-existing challenges to mental health care in under-resourced communities, ongoing conflict presents a host of additional challenges to trauma healing.”

The nod to group psychotherapy is useful, but also lends the question as to why a group-based psychotherapy intervention wouldn’t be the logical next step vs an individual-based trauma therapy like CETA.

7. Overall, the original supposition of the article, that common elements/factors are important and, trauma-focused or not, the intervention helps folks because of common elements/factors, remains unclear since it was not analyzed specifically for this study. It would be ideal to discuss other critical next steps, such as more in-depth quant or qual analysis into the impactful and effective components of each intervention.  

Reviewer 2: 

I thank the authors for their attention to addressing the reviewer comments. There are still some areas that I recommend be improved prior to publication:

1. The details of the sample are a bit conflicting:

- One abstract says 298 participants, the other says 290.

- In the methods it says “The initial pool of participants was 960 adults who lived in Bangui” and then later it says “The initial testing involved 1103 individuals. The first set of participants (N = 143) were used as pilot subjects for the assessors in order to clear up any questions about testing procedures and meaning of the questions. The formal testing was then conducted on 960 participants.” I suggest starting with the 1103 and then explaining how you reduce down to 960. Why was the sample size of 143 selected?

- Why were only 450 participants called out of the 502 who met the HSCL cutoff of 1.75? What happened to the other 52 people?

- Sometimes the authors state that 650 people met the HSCL but off, and other times it says that 502 met the cut-off. Which is correct? Please provide a participant flowchart starting with the 1103 and ending with the 290 indicating how many participants dropped out or were removed from the study at each step.

2. The mechanisms underlying the two interventions are still not well-fleshed out. The HROC intervention is defined as having a “trauma-focus” but the outcome is anxiety and depression symptoms. Please describe the mechanisms of the HROC intervention that are theorized to be the “active ingredients” of anxiety and depression symptoms and why these factors are not theorized to be present in the PVE intervention.

3. At a few points the HSCL is referred to as HSRC. Please correct and spell out HSCL the first time it used.

4. The authors state “The final number of participants was 290, defined as people who participated in at least 3 of the 4 assessment sessions.” How many participants participated in each workshop and each assessment session?

5. Were there any differences in the key variables and demographics of those who were initially screened, those who met inclusion criteria but did not complete the sessions, and those who completed the full study and were included in the final sample of 290 participants?

6. Why were participants not scheduled for the next assessment if they missed the preceding workshop? This process does not align with best practices of an intent-to-treat approach.

7. The decision to use and anxiety/depression measure (the HSCL-25) as the screening criteria for a PTSD intervention is still unclear. There is no literature cited to support the statement “however the HSCL-25 has been used more internationally and is easier to understand compared to the PCL-5.” In addition, the Barbano et al 2019 citation was not listed in the reference list so I could not check the cited reference that the HSCL “targets two major inherent components of PTSD.”

8. The justification for sample size of “we wanted to test as many subjects as possible” is not adequate. Please provide more information to justify the sample size.

9. The authors note “This study was part of a larger effort to de-escalate the violence gripping CAR. The CIPP was an interfaith group composed of Catholic Relief Services (CRS), Aegis Trust (AT), Islamic Relief Worldwide (IRW), and World Vision International (WVI). The function of the group included three major foci: 1. Civil Institutions Establish Leadership Role in Peacebuilding; 2. Livelihoods Security is Re-established; 3. Social Cohesion Fostered through HROC and PVE.” It is not clear what CIPP is or how these organizations interfaced with the study.

10. There are repeated and redundant sections throughout the manuscript.

Reviewer 1 has expressed concerns regarding the broader scientific goals of the manuscript, particularly around the conceptual framing of the hypothesis and the lack of engagement with recent research and broader perspectives in the discussion. Specifically, they noted that some critical comments regarding the article’s scientific goals were not adequately addressed, and that the manuscript could benefit from a more in-depth exploration of the future research directions in the field.

Additionally, Reviewer 1 highlighted the potential bias in the citation patterns, particularly in the discussion, where many of the recent references come from the same high-income country (HIC) institution. In this context, Reviewer 1 has suggested that the authors more explicitly engage with a wider range of sources, including from LMIC contexts, to avoid a narrow perspective and to foster equity in the framing of future research directions.

As editors, we are open to these important suggestions. We encourage the authors to consider how they might revise the manuscript to address Reviewer 1’s concerns by:

Expanding the discussion of the study’s scientific goals, particularly with regard to the field of global mental health and the role of common factors in interventions. This might involve incorporating more recent literature and clearer explanations of how this research fits into broader trends and gaps in the field.

Reassessing the citation patterns in the manuscript to ensure a broader and more equitable representation of research from both HIC and LMIC contexts. This would contribute to a more balanced perspective and address concerns about bias in the literature cited.

Considering ways in which future research in this area can better integrate ethical partnerships between HIC institutions and LMIC organizations, ensuring that research and publications reflect more equitable and inclusive collaborations.

We believe these revisions will help strengthen the manuscript and its relevance to a wider, global audience. We encourage the authors to engage thoughtfully with these suggestions, as doing so will not only enhance the article but also contribute to a richer, more diverse discourse in global mental health research.

We hope that this feedback is helpful as you move forward with revising your manuscript. If you have any questions or require further clarification regarding the reviewers' comments, please don’t hesitate to reach out. We look forward to receiving your revised manuscript and accompanying responses to the reviewers.

Best regards,

Sara Romero